# Validation of plasma microRNAs as biomarkers in sepsis associated acute kidney injury upon first clinical presentation reveals limited diagnostic and prognostic performance

Tamar J. van der Aart[1,2]*, Matthijs Luxen[3,4], Jacqueline Koeze[4,5], Marco van Londen[6], Matthias Hackl[7], Jan C. ter Maaten[1,2], Matijs van Meurs[4,5], Hjalmar R. Bouma[1,8], on behalf of the the Acutelines research group¶

1 Department of Internal Medicine, University Medical Center Groningen, University of Groningen, Groningen, The Netherlands, 2 Department of Acute Care, University Medical Center Groningen, University of Groningen, Groningen, The Netherlands, 3 Department of Pathology and Medical Biology, Medical Biology section, University Medical Center Groningen, University of Groningen, Groningen, The Netherlands, 4 Department of Critical Care, University Medical Center Groningen, University of Groningen, Groningen, The Netherlands, 5 Healics study group, Department of Critical Care, University Medical Center Groningen, University of Groningen, Groningen, The Netherlands, 6 Department of Internal Medicine, Division of Nephrology, University Medical Center Groningen, University of Groningen, Groningen, The Netherlands, 7 TAmiRNA GmbH, Vienna, Austria, 8 Department of Clinical Pharmacy and Pharmacology, University Medical Center Groningen, University of Groningen, Groningen, The Netherlands

¶ Membership of the Acutelines research group can be found at www.acutelines.nl
* t.j.van.der.aart@umcg.nl

## Abstract

### Background

Sepsis is a life-threatening response to an infection, often complicated by sepsis-associated acute kidney injury (SA-AKI). Early recognition of SA-AKI is critical but challenged by the limited sensitivity of existing diagnostic markers. MicroRNAs (miRNAs), which regulate key SA-AKI pathways, have shown diagnostic promise, yet their clinical utility in early SA-AKI recognition remains unexplored. Moreover, validation in relevant clinical settings and populations remains limited. Therefore, this study aims to explore the potential of miRNAs for early recognition of SA-AKI at emergency department (ED) presentation, and explore the generalizability of findings by including a cohort of intensive care urine (ICU) patients with more advanced disease.

### Methods

We conducted a post-hoc analysis of prospectively collected data from patients admitted to the ED and ICU. We performed a thorough literature review to select twelve miRNAs, previously implicated in kidney injury and sepsis or SA-AKI. MiRNAs were extracted from plasma and quantified using qPCR with and normalization to the global mean. We measured plasma levels of selected miRNAs upon ED arrival in 193

**Data availability statement:** All relevant data are within the manuscript and its Supporting Information files.

**Funding:** Acutelines is cofounded by the UMCG TJ van der Aart is supported by an MD PhD grant from the University of Groningen. The funders had no role in study design, data collection and analysis, decision to publish, or preparation of the manuscript.

**Competing interests:** MH is co-founder, shareholder, and employee of TAmiRNA GmbH Measuring and quantifying microRNA expression was conducted at TAmiRNA GmbH.

**Abbreviations:** AKI, Acute Kidney Injury; APACHE, Acute Physiology and Chronic Health Evaluation; CKD, Chronic Kidney Disease; ED, Emergency Department; ICU, Intensive Care Unit; KDIGO, Kidney Disease: Improving Global Outcomes; miRNA, MicroRNA; NEWS, National Early Warning Score; SA-AKI, Sepsis-associated Acute Kidney Injury; SOFA, Sequential Organ Failure Assessment.

acutely ill patients (no infection, n = 65; sepsis, n = 67; SA-AKI, n = 61), and 47 critically ill patients (sepsis, n = 18; SA-AKI, n = 29). Statistical analyses included logistic and Cox regression adjusted for clinical variables, with correlations assessed between miRNA levels and disease severity markers. Diagnostic performance was evaluated using receiver operating characteristic (ROC) curve analysis.

## Results

MiR-21-5p (OR 2.28, 95% CI [1.40–3.73]; p < 0.01) and miR-16-5p (OR 0.74, 95% CI [0.59–0.93]; p = 0.01) levels were associated with SA-AKI at ED presentation. Furthermore, miR-21-5p was independently associated with 30-day mortality after adjusting for age, illness severity, and comorbidities (adjusted OR 2.30, 95% CI [1.38–3.86]; p < 0.01). Similarly, in the ICU cohort with more advanced sepsis, miR-21-5p was associated with SA-AKI (OR 3.48, 95% CI [1.27–9.53]; P = 0.02), achieving an AUC of 0.74 (95% CI [0.58–0.89]), although it was not associated with 30-day mortality in this cohort.

## Conclusion

We selected twelve miRNAs through literature review associated with kidney injury, sepsis or SA-AKI. Of these, only miR-21-5p was associated with SA-AKI and predicted 30-day mortality upon ED admission. This analysis effectively serves as a negative validation for most literature-derived miRNAs, challenging their clinical applicability identification of SA-AKI both at early presentation and in more advanced stages.

## Trial registration

This study is embedded in the Acutelines data-biobank (www.acutelines.nl), registered in Clinicaltrials.gov (NCT04615065, November 3rd 2020) and the The Biobank Intensive Care Groningen registered in Clinicaltrials.gov (NCT04502511, August 6th 2020).

## Introduction

Sepsis is a dysregulated host response to an infection often complicated by acute kidney injury, referred to as sepsis-associated acute kidney injury (SA-AKI). The development of AKI in sepsis is strongly associated with failure of other organs and increased mortality, even in relatively mild disease [1–3]. Despite its severity, up to 40% of sepsis cases are not recognized at the emergency department (ED) [4]. As the primary point of hospital presentation for community-acquired sepsis, the ED plays a critical role in the early recognition of SA-AKI, which is essential for timely risk stratification and resource allocation [5,6]. However, current classifications of AKI, based on serum creatinine levels or urine output, lack specificity and fail to capture the full scope or nature of renal injury [7,8]. These classifications rely on changes from baseline kidney function, which is frequently unknown or hard to establish in the

ED due to lack of historical kidney data [8–11]. As a result, early diagnosis and accurate risk stratification remain challenging. Reliable markers are needed that facilitate recognition of SA-AKI at the ED, ultimately improving outcomes for patients.

MicroRNAs (miRNAs) are short non-coding RNAs which are involved in modulating gene expression [12–15]. miRNAs regulate pathways involved in the pathophysiology of SA-AKI, including inflammation, oxidative stress, apoptosis, and tubular epithelial injury [16–19]. Therefore, their potential as biomarker for SA-AKI and as mediator of disease has been explored by multiple studies [20–24]. In addition to improving early recognition, increased understanding of their role in the pathophysiology of SA-AKI could reveal new therapeutic strategies [12,25–27]. Despite multiple studies exploring the role of miRNAs in SA-AKI, their clinical applicability remains uncertain. Existing studies exhibit substantial variability in study design, patient populations, and endpoints, complicating direct comparison. Furthermore, many studies focus on single miRNAs, with conflicting findings regarding their diagnostic and prognostic value. To address these inconsistencies, we conducted a literature review to select a panel of miRNAs previously associated with sepsis and kidney injury or SA-AKI, aiming to evaluate their applicability in clinical settings. These were subsequently validated in a well-defined cohort comprising both ED and ICU patients, allowing for the evaluation of circulating miRNA levels in patients at varying sepsis stages. This approach enabled us to systematically assess the diagnostic and prognostic potential of these miRNAs in SA-AKI.

## Methods

### Study design

This study is a post-hoc analysis of prospectively collected data, including miRNA data from human plasma samples collected at ED and ICU admission. This study is embedded in the Acutelines data-biobank (www.acutelines.nl), registered in Clinicaltrials.gov (NCT04615065, November 3rd 2020) and The Biobank Intensive Care Groningen registered in Clinicaltrials.gov (NCT04502511, August 6th 2020) hereafter referred to as Biobank IC [28]. The aim of the study was to investigate and directly compare the potential of different miRNAs previously associated with sepsis and kidney injury or SA-AKI, where differences in study design, patient populations (ED vs. ICU), sample types and endpoints across prior studies precludes direct comparisons but also undermines the clinical applicability of existing findings. Following a comprehensive literature review, we identified twelve miRNAs associated with sepsis and kidney injury or SA-AKI. These miRNAs were prospectively measured in patients presenting to the ED with sepsis—retrospectively classified for SA-AKI—as well as in a separate cohort of ICU patients with more advanced sepsis. This study was approved by the medical ethics committee of the University Medical Centre Groningen (UMCG), a tertiary care teaching hospital in The Netherlands. All procedures were conducted in accordance with the Helsinki Declaration. Data were pseudomized to ensure confidentiality and privacy in accordance with European privacy legislation.

### miRNA selection

The selection of miRNA candidates as potential biomarkers was based on a comprehensive review of 116 research articles focused on sepsis, SA-AKI, miRNA expression in the kidney, kidney injury or AKI (S1 Table–S3 Table, S1 Fig and S2 Fig). These articles included reviews, experimental research, and clinical studies, covering miRNA investigations across various sample types (liquid biopsies and tissue) and species (human and animal models) (S1 Table–S3 Table, S1 Fig and S2 Fig). We selected miRNAs that were consistently associated with either sepsis and kidney injury or SA-AKI across multiple studies and were supported by both experimental studies and clinical research studies in humans (S1 Table–S3 Table, S1 Fig and S2 Fig) This approach ensured that the selected miRNAs were grounded in evidence from diverse sources, enhancing their potential to serve as a relevant panel for the early detection and stratification of SA-AKI.

## Participants

This study includes data from patients presenting at the ED with sepsis and SA-AKI as well as a control group consisting of non-infection. All patients admitted to the ED from February 2021 until February 2023 who provided plasma and written informed consent, where applicable by proxy, for participation in the Acutelines cohort were screened for participation. Acutelines is a multi-disciplinary prospective hospital-based cohort study at the ED of the UMCG. Acutelines' complete protocol and overview of the current, full data dictionary is available via www.acutelines.nl [29].

Additionally, we investigated an ICU cohort consisting of severely ill patients with sepsis and SA-AKI necessitating organ support therapy. Patients participating in the cohort Biobank IC admitted to the ICU between December 2021 and March 2023 with severe sepsis were screened for inclusion. Participants of Biobank IC are retrospectively asked for written informed consent, where applicable by proxy, within three months after ICU admission. Of note, within the Dutch healthcare system, ICU admission is reserved for severe illnesses requiring organ support. Sepsis cases not requiring invasive respiratory support, vasopressors, or renal replacement therapy (RRT) are treated on the general wards.

## Definitions

Sepsis was defined according to the sepsis-3 criteria as an increase in SOFA score of two or more within 48 hours of ED presentation. The presence and site of infection were post hoc determined by an expert adjudication committee investigating the complete medical file as part of Acutelines, based on the Centers for Disease Control and Prevention consensus definitions [30]. AKI was determined by the Kidney Disease: Improving Global Outcomes (KDIGO) criteria: an increase in creatinine of at least 26.5 µmol/L or an increase to at least 1.5 times the baseline creatinine level [31]. The baseline creatinine level in this study was calculated as the mean plasma creatinine from pre-existing plasma creatinine measurements measured up to 12 months prior to ED presentation [10]. The control group consisted of patients who presented to the ED with suspected infection, as indicated by the treating physician, but who were ultimately determined not to have an infection based on post hoc adjudication.

In accordance with the Dutch National Intensive Care Evaluation (NICE) guidelines, sepsis in the ICU cohort is defined as an infection requiring ICU admission for organ supportive therapy [32]. Presence and site of infection was determined by either positive culture results and/or radiological imaging combined with clinical signs of infection. AKI was determined through the NICE guidelines: the necessity of renal replacement therapy (RRT) within the first 24 hours of ICU admission or a plasma creatinine level of ≥300 µmol/L in the past 24 hours accompanied by oliguria [33]. Oliguria was defined as a urine output of less than or equal to 150 ml over eight consecutive hours without other causes.

## Data collection

Medical data collected primarily for research purposes is collected and managed using REDCap (Vanderbilt University, Nashville, USA) electronic data capture tools hosted at the UMCG [34,35]. All medical data, including vital parameters and laboratory results, is obtained from the electronic health records of the hospital (EPIC systems, Verona, USA). Comorbidities are based on the Charlson Comorbidity Index and obtained from the electronic medical file for each patient [36].

## Plasma collection

Blood samples were collected within four hours upon ED arrival in an EDTA Sarstedt tube (BD Vacutainer® PPT™, Franklin Lakes, USA). Blood samples were centrifuged within 2–4 hours after ED admission (20°C, 1500 g, 15 min), and stored at –80°C. All samples underwent a single freeze-thaw cycle. Blood samples at the ICU were collected within three hours of ICU admission in an EDTA Sarstedt tube, centrifugated (4°C, 380 g, 15 min), and stored at –80°C.

## miRNA extraction and analysis

miRNAs were analyzed in the plasma of a total of 240 patients across two cohorts. Total RNA was extracted from 200 µL human plasma using the Maxwell RSC miRNA Tissue kit (Promega, Madison, USA), performed at the laboratory of

TAmiRNA (TAmiRNA GmbH, Vienna, Austria). Total RNA was eluted from beads in 50 μL nuclease-free water and stored at −80°C. Equal volumes of total RNA were then utilized for reverse transcription using the miRCURY LNA RT kit (Qiagen, Hilden, Germany) according to the manufacturer's instructions. qPCR was performed using miRCURY LNA microRNA PCR assays (Qiagen) on LightCycler 480 II instrument (Roche, Bazel, Switzerland). Cq-values were calculated on the Roche software using the second-derivate maximum method, and normalized to the global mean (GM) by averaging all measured miRNAs except miR-127, which was excluded due to missing data (S3 Fig) [37]. Normalization was conducted using the following equation:

$$(\Delta - Cq) = (Cq\ microRNA - Global\ Mean) \times -1$$

## Statistical analysis

All statistical analyses were performed using R (Version 4.3.1, R Core Team 2021, Vienna, Austria). Baseline characteristics were analyzed using a combination of non-parametric and parametric tests, including one-way ANOVA or t test for normally distributed data and a Kruskal Wallis test or Mann Whitney U test for non-normally distributed data. A post hoc Dunn test and post hoc Tukey test with Bonferroni adjustment were used, when applicable. An Anderson-Darling test was used to determine normal distribution. Associations with 30-day mortality were investigated with logistic regression and a multivariable COX regression adjusted for age, comorbidities and disease severity expressed as national early warning score (NEWS2) or Acute Physiology And Chronic Health Evaluation score (APACHE IV). Results were visualized using Receiver Operating Characteristic (ROC) curve analyses. High miRNA levels were defined as values exceeding the median level of the respective miRNA. A Spearman's rank correlation analysis was conducted between miRNA levels and AKI severity, disease severity scores and laboratory parameters, including C-reactive protein (CRP), leukocyte count, lactate, plasma creatinine, estimated glomerular filtration rate (eGFR), bilirubin, and plasma urea concentration. Statistical significance was determined at $p < 0.05$ throughout.

# Results

## miRNA selection

We reviewed studies that investigated miRNAs in the context of sepsis, kidney injury and SA-AKI, prioritizing those that included comparable patient populations, employed similar endpoints, and provided both clinical and experimental evidence linking specific miRNAs to inflammatory, apoptotic, and renal injury pathways. Our literature review revealed 116 research papers describing miRNAs in relation to sepsis and kidney injury or SA-AKI (S1 Table–S3 Table, S1 Fig and S2 Fig) Based on review of these research papers, we identified twelve miRNAs associated with SA-AKI for validation in our cohort.

## Characteristics of the ED population

We compared plasma levels of twelve literature-derived miRNAs in 193 patients at ED presentation, categorized into three groups: 65 (33.7%) non-infectious controls, 67 (34.7%) patients with sepsis, and 61 (31.6%) patients with SA-AKI (Table 1). Patients with sepsis (median age 67, IQR [47–70]) and SA-AKI (median age 69, IQR [53–77]) were on average older than non-infection patients (median age 61, IQR [47–70]; $p = 0.02$). Comorbidities were most prevalent among SA-AKI patients and included diabetes mellitus ($p < 0.01$), cardiovascular disease (CVD, $P = 0.03$), malignancy ($P < 0.01$), COPD ($p < 0.01$), and chronic kidney disease (CKD, $p < 0.01$). Pulmonary infections were the most common source of infection in both sepsis (43%) and SA-AKI (31%), followed by urogenital (19% and 16%, respectively) and abdominal infections (9% and 11%, respectively) (S4 Table). We observed worse clinical outcomes in patients with SA-AKI compared to septic patients and the control group, represented by the longest hospital stay (median 8 days in SA-AKI vs. 5 days in sepsis and 2 days in

**Table 1. Baseline characteristics of both the emergency department and intensive care population (n = 240). Statistical analysis includes both parametric and non-parametric tests: one-way ANOVA or t-test for normally distributed data, and Kruskal-Wallis test or Mann-Whitney U test for non-normally distributed data. Post hoc comparisons were conducted using the Dunn test (for Kruskal-Wallis) and Tukey test with Bonferroni adjustment (for ANOVA), when applicable. Normality was assessed using the Anderson-Darling test. Data are presented as mean (SD), n (%), or median [IQR]. Cardiovascular Disease (CVD); Chronic Obstructive Pulmonary Disease (COPD), Chronic Kidney Disease (CKD).**

| | Emergency department cohort | | | | Intensive care unit cohort | | |
|---|---|---|---|---|---|---|---|
| | No infection (N = 65) | Sepsis (N = 67) | SA-AKI (N = 61) | P value | Sepsis (N = 18) | SA-AKI (N = 29) | P value |
| **Baseline characteristics** | | | | | | | |
| Age | 61 [47-70] | 67 [53-75] | 69 [53-77] | **0.02** | 60 [52-67] | 68 [59-71] | 0.07 |
| Sex (F) | 32 (49) | 32 (48) | 31 (51) | 0.94 | 8 (44) | 14 (48) | 0.87 |
| **Disease severity scores** | | | | <0.01 | | | |
| NEWS2 | 2 [0-4] | 7 [4-9] | 5 [3-9] | <0.01 | | | |
| SOFA | 1 [0-4] | 5 [4-6] | 7 [5-9] | | | | |
| APACHE | | | | | 92 [49-102] | 68 [48-117] | 0.65 |
| **AKI severity** | | | | | | | |
| **AKI G1** | | | 32 (52) | | | | |
| **AKI G2/G3** | | | 29 (48) | | | 18 (100) | |
| **Laboratory parameters** | | | | | | | |
| Baseline creatinine µmol/L | 75 [58-85] | 70 [58-85] | 75 [65-101] | 0.29 | | | |
| Creatinine µmol/L | 75 [65-83] | 73 [55-92] | 133 [110-181] | **<0.01** | 178 [88-310] | 150 [90-211] | 0.47 |
| Urea mmol/L | 5.4 [4.3-7.0] | 5.8 [4.2-7.4] | 12.5 [7.3-17.1] | **<0.01** | 11.2 [6.1-14.4] | 9.2 [6.6-13.4] | 0.89 |
| eGFR ml/min/1.73m² | 88 [78-102] | 87 [64-102] | 41 [26-54] | **<0.01** | 94 [73-102] | 30 [20-48] | **<0.01** |
| Hemoglobin mmol/L | 8.1 (1.6) | 8.0 (1.3) | 7.3 (1.3) | **<0.01** | 7.1 (1.5) | 6.5 (1.6) | 0.31 |
| Thrombocytes x10⁹/L | 243 [205-302] | 208 [140-264] | 217 [151-286] | **0.03** | 210 [89-262] | 166 [78-265] | 0.67 |
| Leukocytes x10⁹/L | 7.9 [6.6-11.0] | 11.7 [7.0-15.5] | 12.4 [7.2-18.7] | **<0.01** | 8.9 [4.4-17.8] | 12.4 [4.9-19.3] | 0.65 |
| CRP mg/L | 19 [6-41] | 90 [39-175] | 167 [80-257] | **<0.01** | 161 [129-320] | 170 [70-320] | 0.79 |
| Lactate mmol/L | 1.7 [1.0-2.2] | 1.4 [1.0-2.3] | 2.0 [1.4-3.5] | **<0.01** | 1.6 [1.3-2.2] | 2.0 [1.2-4.1] | 0.41 |
| **Comorbidities** | | | | | | | |
| Diabetes mellitus | 5 (8) | 9 (13) | 20 (33) | <0.01 | 3 (17) | 5 (17) | 0.96 |
| CVD | 7 (11) | 16 (24) | 20 (30) | 0.03 | 1 (6) | 3 (10) | 0.57 |
| Malignancy | 3 (5) | 2 (3) | 26 (42) | <0.01 | 4 (22) | 4 (14) | 0.46 |
| COPD | 1 (2) | 24 (36) | 13 (21) | <0.01 | 1 (6) | 3 (10) | 0.57 |
| CKD | 0 | 1 (1) | 12 (20) | <0.01 | 2 (11) | 2 (7) | 1 |
| **Diagnosis admission** | | | | | | | |
| Infection/Sepsis | 0 (0) | 67 (100) | 61 (100) | | 13 (72) | 29 (100) | |
| Cardiac | 9 (14) | | | | 1 (6) [a] | | |
| Neurological | 18 (28) | | | | | | |
| Respiratory | 5 (7) | | | | 1 (6) [b] | | |
| Intoxication | 5 (7) | | | | | | |
| Gastro-intestinal | 12 (19) | | | | | | |
| Anaphylaxis | 5 (8) | | | | | | |
| Other | 11 (17) | | | | 3 (6) [c] | | |
| **Outcome** | | | | | | | |
| Length of hospital stay | 2 [2-5] | 5 [3-7] | 8 [5-16] | **<0.01** | 17 [8-28] | 16 [7-21] | 0.48 |
| ICU admission | 1 (2) | 12 (18) | 20 (33) | 0.01 | | | |
| Length of ICU stay | 7 | 2 [2-3] | 4 [2-6] | 0.15 | 4 [3-9] | 4 [2-8] | 0.89 |
| 30-day mortality | 0 | 9 (13) | 15 (25) | <0.01 | 7 (39) | 12 (41) | 0.87 |

[a]Cardiac arrest complicated by pulmonary sepsis

[b]Obstructed airway complicated by pulmonary sepsis

[c]Acid base disturbance complicated by pulmonary sepsis, ruptured aneurysm complicated by infected endovascular prothesis, elective orthopaedic surgery complicated by infected prothesis

controls, p<0.01), the highest ICU admission rate (33% in SA-AKI vs. 18% in sepsis and 2% in controls, p=0.01) and the highest 30-day mortality rate (25% in SA-AKI vs 13% in sepsis and 0% in controls, p<0.01).

## Few miRNAs differ upon presentation to the ED

Of all studied miRNAs, only miR-10a-5p (p=0.02), miR-16-5p (p=0.03), and miR-21-5p (p<0.01, Fig 1) were different between the three groups. miR-10a-5p levels were lower in both sepsis (median −7.94, IQR [−9.00 to −7.14]) and SA-AKI (median −7.88, IQR [−8.63 to −7.18]) compared to non-infection (median −7.37, IQR [−8.16 to −6.86]; p=0.04), but did not differ between sepsis and SA-AKI. miR-16-5p levels were lower in SA-AKI (mean 6.91 [SD 1.61]) compared to both sepsis (mean 7.51 [1.36]; p=0.03) and non-infection (mean 7.46 [1.19]; p=0.03). miR-21-5p levels were higher in SA-AKI (median 5.74, IQR [5.32–6.33]) compared to sepsis (median 5.51, IQR [5.12–5.96]; p=0.04) and non-infection (median 5.37, IQR [5.17–5.69]; p<0.01). In conclusion, at the time of ED presentation, only the levels of three miRNAs differed for SA-AKI.

## miRNAs as biomarkers for early recognition of SA-AKI at the ED

To evaluate whether plasma miRNA levels could aid in the identification of SA-AKI at ED presentation, we conducted a logistic regression analysis. From the twelve miRNAs, only miR-21-5p (OR 2.28, 95% CI [1.40–3.73]; p<0.01) and miR-16-5p (OR 0.74, 95% CI [0.59–0.93]; p=0.01) were associated with SA-AKI (Table 2). miR-10a-5p levels were not associated with SA-AKI (p=0.28). The combined diagnostic performance of miR-21-5p and miR-16-5p in a multivariable model demonstrated an AUC of 0.68 (95% CI [0.60–0.76]) with a good model fit (Hosmer-Lemeshow p=0.89) (Fig 2).

## miR-21-5p and miR-16-5p are associated with AKI severity at the ED

We explored the relationship between these miRNAs and AKI severity according to the KDIGO criteria [31]. miR-21-5p levels increased with increased AKI severity and were higher in AKI KDIGO grade 1 compared to grade 3 (p=0.02) (Fig 3). In contrast, miR-16-5p levels decreased with AKI severity and were lower in patients with AKI KDIGO grade 3 compared to those both grade 1 and grade 2 (p<0.01 and p<0.01, respectively). miR-21-5p correlated negatively with eGFR (r=−0.17, p=0.02) and positively with urea (r=0.26, p<0.01) and creatinine (r=0.15, p=0.04) (S1 Fig). There were no strong correlations between the measured miRNAs and clinical parameters, including illness severity scores, laboratory parameters of inflammation or organ dysfunction (S4 Fig).

## Elevated miR-21-5p levels are an independent risk factor for short-term mortality among ED patients

Given previous reports linking the selected miRNAs to clinical outcomes, we evaluated their ability to predict 30-day mortality in our cohort. Among the evaluated miRNAs, miR-21-5p (OR 3.43, 95% CI [1.70–6.55]) and miR-16-5p (OR 0.70, 95% CI [0.51–0.96]) were associated with 30-day mortality (S5 Table). We performed a multivariable Cox regression analysis to assess the independent association of these miRNA levels, adjusting for age, sex, comorbidities, and disease severity as defined by the NEWS2 score. Notably, miR-21-5p emerged as an independent predictor of 30-day mortality (adjusted OR 2.30, 95% CI [1.38–3.86], p<0.01), with an AUC of 0.76 (95% CI: 0.65–0.87) (Table 3). In contrast, miR-16-5p was not independently associated with 30-day mortality after adjusting for age, sex, comorbidities, and disease severity (p=0.18).

## Validation of miRNAs in critically ill patients with sepsis

Next, we validated the selected miRNAs in a more severely ill ICU cohort to explore the generalizability of their diagnostic and prognostic value across different stages of sepsis severity. The ICU cohort consisted of patients with severe sepsis requiring organ support: 18 with sepsis (38%) and 29 with SA-AKI (62%) (Table 1). Patients with SA-AKI were older than those with sepsis (median age 68 [59–71] vs. 60 years [52–67]; p=0.07). Similar to the in the ED cohort, pulmonary infections were the most common in both sepsis and SA-AKI (28% and 41%, respectively), followed by abdominal tract

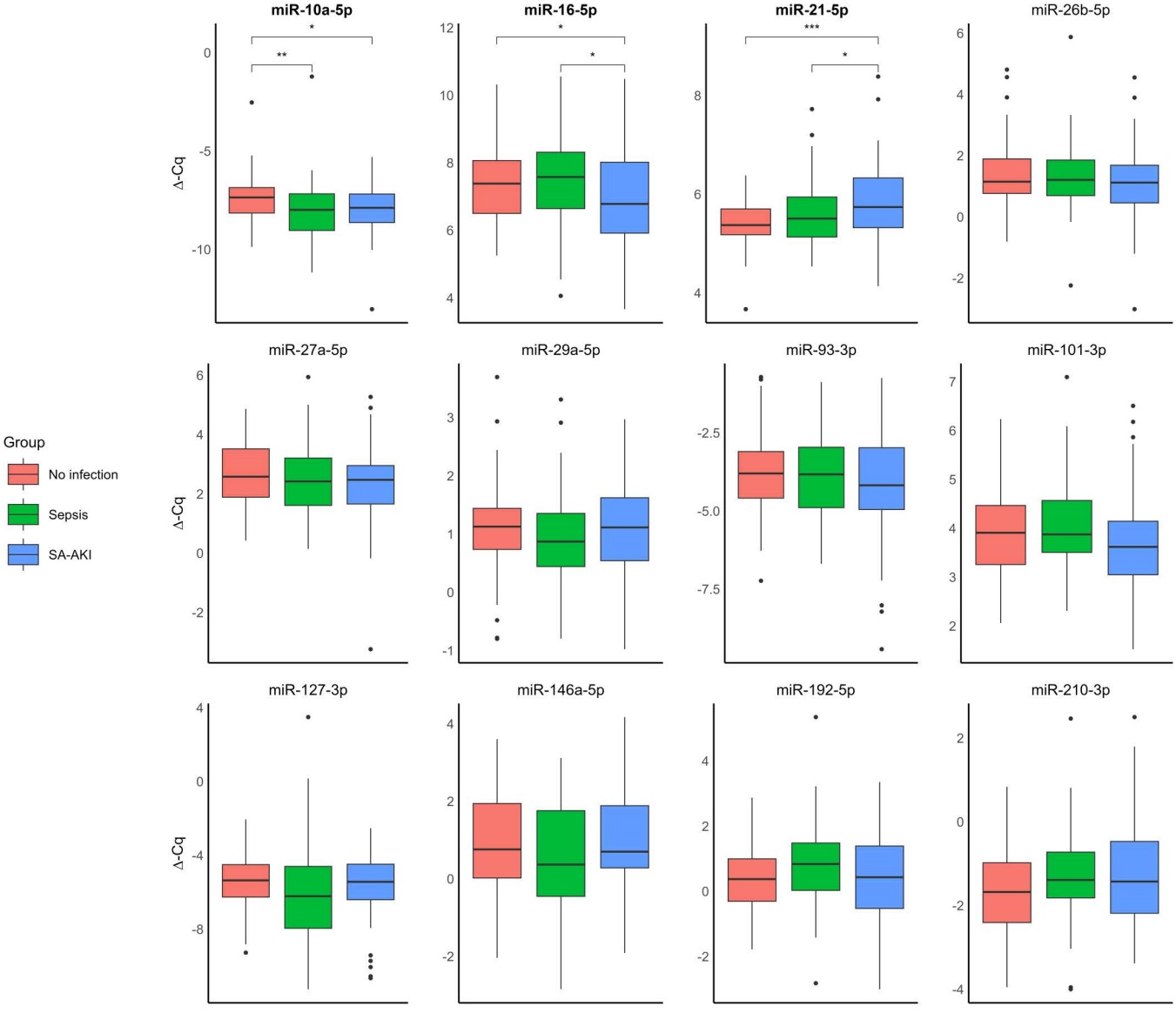

**Fig 1. Comparison of relative expression patterns of microRNAs between patients without infection at the emergency department (No infection, red), patients with sepsis at the ED (Sepsis, green), and patients with sepsis-associated acute kidney injury at the ED (SA-AKI, blue).** ANOVA and Kruskal-Wallis test were used for statistical testing of normally respective non-normally distributed data. Significant differences are depicted by * P < 0.05, ** P < 0.005, *** P < 0.001 and titles are **bold**.

infections (22% and 17%, respectively) (S4 Table). Disease severity, as measured by the APACHE IV score (median 92 and 68, p = 0.65), and 30-day mortality rates were comparable between sepsis and SA-AKI (39% and 41% (p = 0.90).

## miR-21-5p is elevated in critically ill patients with sepsis

Only miR-21-5p exhibited altered plasma levels in ICU patients, with higher levels in SA-AKI compared to sepsis (median 5.53 vs 5.93; p < 0.01) (Fig 4). miR-21-5p was associated with SA-AKI (OR 3.48, 95% CI [1.27–9.53]; p = 0.02), with an AUC of 0.74 (95% CI 0.59–0.89; p = 0.02) (Table 4). However, unlike in the ED cohort, miR-21-5p was not associated with 30-day mortality (S6–S7 Table). None of the other miRNAs were associated with SA-AKI or 30-day mortality.

**Table 2. Association of circulating microRNA levels from plasma with sepsis-associated acute kidney injury (SA-AKI) in an emergency department population.** Odds ratio (OR), and area under the receiver operator curve (AUROC) with their respective 95% confidence intervals (CI) are presented in the table below. Colors correspond to the lines in **Fig 2**. Bold indicates significant association.

| microRNA | OR (95% CI) | AUC (95% CI) |
|---|---|---|
| miR-16-5p + miR-21-5p | | 0.68 (0.60-0.76) |
| miR-10a-5p | 0.88 (0.69–1.11) | 0.54 (0.44-0.63) |
| **miR-16-5p** | **0.74 (0.59–0.93)** | **0.61 (0.52-0.71)** |
| **miR-21-5p** | **2.28 (1.40–3.73)** | **0.65 (0.56-0.74)** |
| miR-26b-5p | 0.76 (0.57–1.02) | 0.57 (0.48-0.65) |
| miR-27a-5p | 0.90 (0.69–1.16) | 0.53 (0.44-0.61) |
| miR-29a-5p | 1.12 (0.75–1.66) | 0.53 (0.44-0.62) |
| miR-93-3p | 0.83 (0.66–1.03) | 0.55 (0.45-0.66) |
| miR-101-3p | 0.77 (0.56–1.05) | 0.58 (0.49-0.67) |
| miR-127-3p | 1.01 (0.85 - 1.21) | 0.55 (0.44-0.66) |
| miR-146a-5p | 1.15 (0.91–1.44) | 0.55 (0.47-0.64) |
| miR-192-5p | 0.86 (0.67–1.11) | 0.54 (0.45-0.63) |
| miR-210-3p | 1.15 (0.88–1.50) | 0.53 (0.43-0.62) |

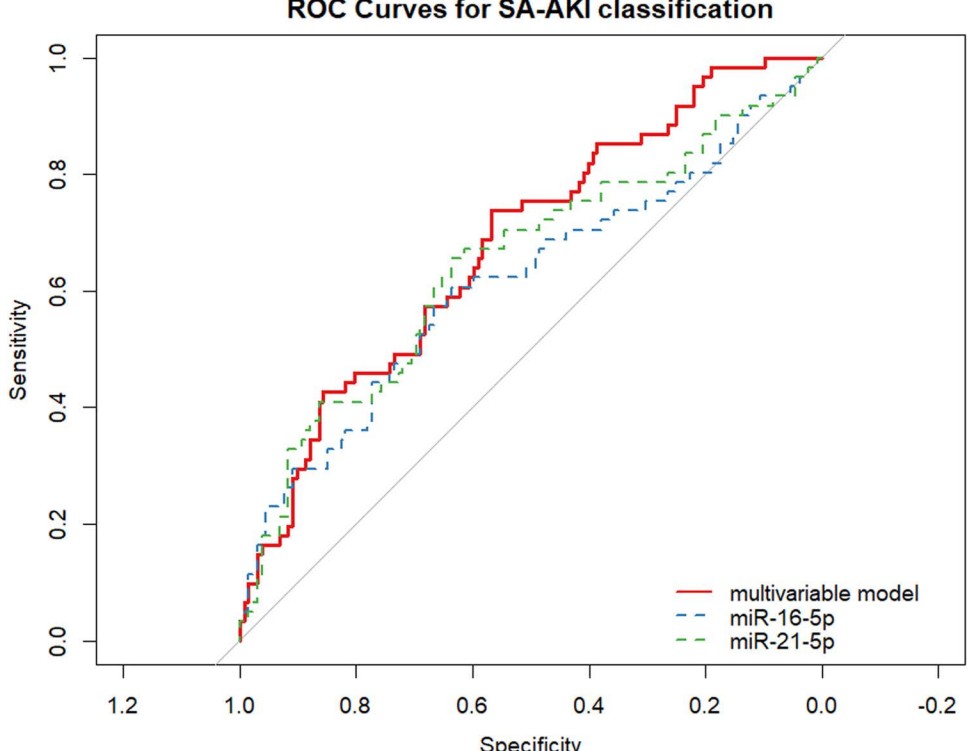

**Fig 2. Receiver Operating Characteristic (ROC) curves for the classification of sepsis-associated acute kidney injury (SA-AKI) at presentation to the Emergency Department (ED) based on microRNA level differences.** ROC curves for miR-16-5p (blue) and miR-21-5p (green) are shown. The multivariable logistic model (red) consists of both miR-16-5p and miR-21-5p. The odds ratio (OR), area under the curve (AUC), and their respective 95% confidence intervals (CI) are provided in Table 2.

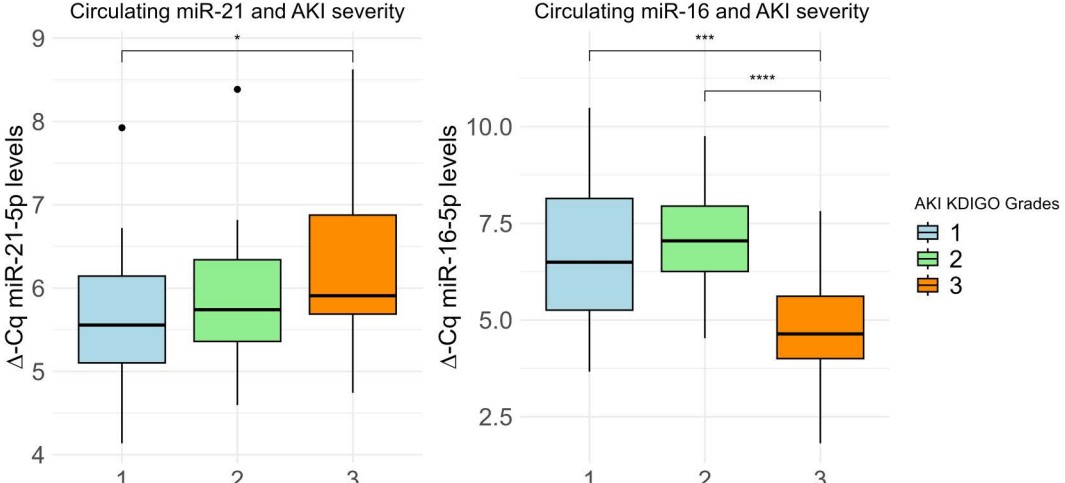

**Fig 3. Boxplots showing the differences in miR-21-5p and miR-16-5p levels across increasing AKI severity based on Kidney Disease: Improving Global Outcomes (KDIGO) grades.** Elevated miR-21-5p levels and decreased miR-16-5p levels correspond to greater AKI severity. KDIGO grade 1 is represented in blue, grade 2 in green, and grade 3 in orange. Kruskal-Wallis test was used for statistical testing. Statistical significance is indicated as $*p < 0.05$, $***p < 0.005$, and $****p < 0.001$.

**Table 3. Association of miR-21-5p with 30-day mortality in a cox regression model adjusted for potential confounders.**

|  | **microRNA** | **HR (95% CI)** |
|---|---|---|
| Crude model | miR-21-5p | **4.26 (2.16–8.38)** |
| Adjusted model | miR-21-5p | **2.30 (1.38–3.86)**[*] |
|  | Age | 1.03 (0.99–1.07) |
|  | COPD | 1.40 (0.55–3.55) |
|  | Diabetes mellitus | 0.99 (0.35–2.75) |
|  | Cardiovascular disease | 1.14 (0.43–3.05) |
|  | Malignancy | 2.23 (0.88–5.65) |
|  | Chronic kidney disease | 1.25 (0.37–4.20) |
|  | NEWS2 score | 1.15 (1.02–1.30) |

[*]$p < 0.01$

## Discussion

In this study, we conducted a thorough literature review which resulted in the selection of twelve miRNAs associated with kidney injury and sepsis or SA-AKI as potential biomarkers for the early identification of SA-AKI (Table 5). To increase the generalizability and robustness of the associations, we aimed not only to evaluate their diagnostic potential in ED patients at the clinical early stages of disease, but also to explore their stability in a separate ICU cohort with more advanced sepsis. We therefore measured plasma levels of all selected miRNAs in 193 patients presenting to the ED with sepsis and SA-AKI and validated these findings in 57 patients with sepsis and SA-AKI admitted to the ICU. Our findings show that increased levels of miR-21-5p and decreased levels of miR-16-5p were significantly associated with SA-AKI at ED presentation, suggesting their potential utility as early biomarkers for SA-AKI. Moreover, miR-21-5p was independently associated with 30-day mortality. To explore whether findings extended to more severe disease, we conducted a secondary,

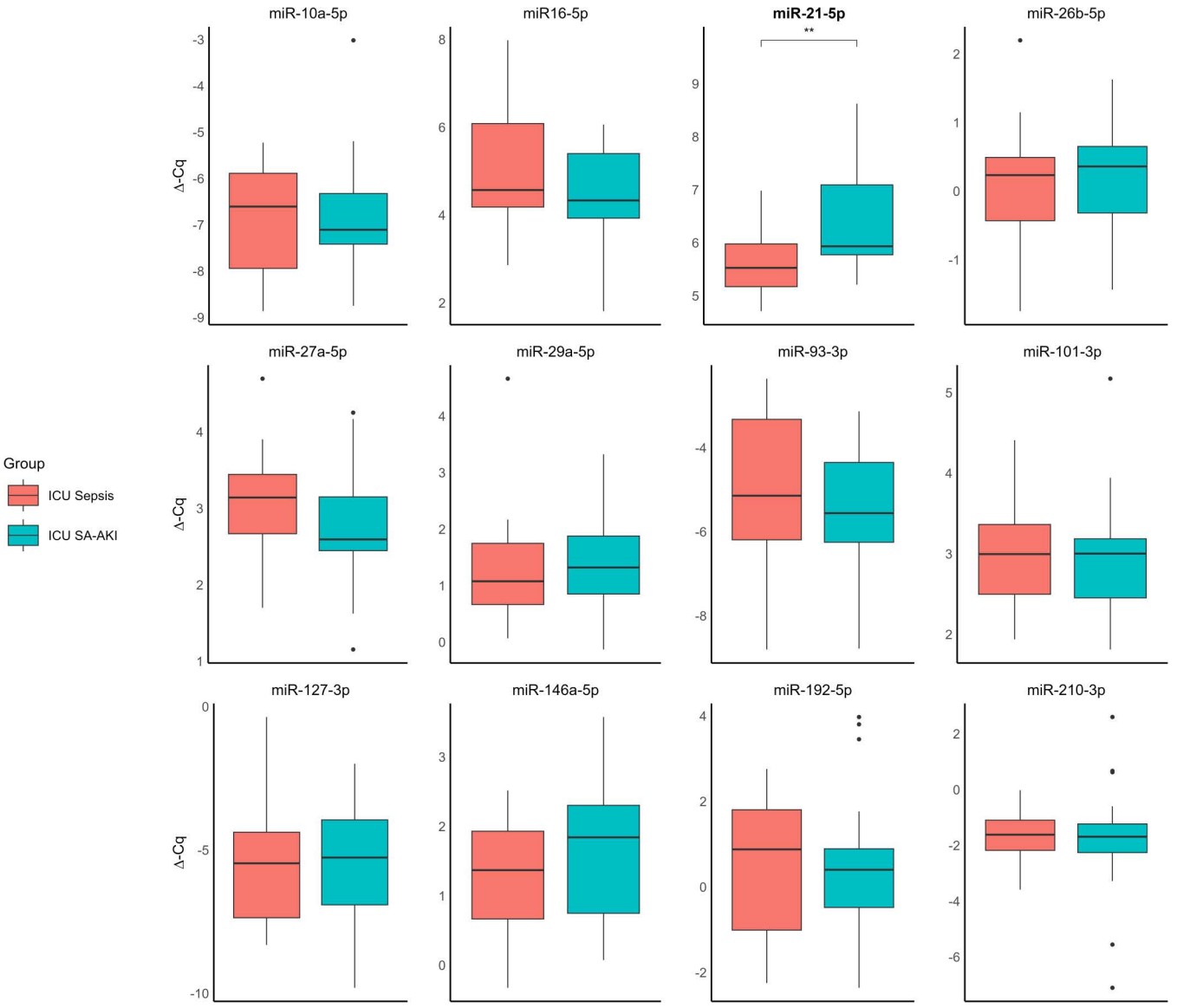

**Fig 4. Comparing microRNA levels between patients with sepsis (ICU Sepsis, Red) and sepsis associated acute kidney injury (ICU SA-AKI, Blue) in the intensive care unit cohort, using t-test for normally distributed microRNA levels or the Mann-Whitney U test for non-normally distributed data.** Significance is depicted by * P < 0.05, ** P < 0.01, with titles in **bold**.

exploratory analysis in ICU sepsis patients. Only miR-21-5p remained associated with SA-AKI, with no observed association with mortality. Taken together, with the exception of miR-21-5p, we were unable to confirm the previously reported associations between the majority of these miRNAs and SA-AKI. These findings raise questions about the clinical applicability of circulating miRNAs as reliable tools for SA-AKI recognition, as well as their association with sepsis and kidney injury.

Early recognition of SA-AKI in patients presenting to the ED remains a clinical challenge due to the absence of rapid and reliable diagnostic tools [4,8,38,39]. Our findings indicate that at ED presentation miR-21-5p plasma levels are elevated and miR-16-5p levels reduced in patients with SA-AKI compared to patients with sepsis and non-infectious patients.

**Table 4. Differentiation of sepsis-associated acute kidney injury (SA-AKI) in the Intensive care unit population based on circulating microRNA levels in plasma. Odds ratio (OR), and area under the receiver operator curve (AUROC) with their respective 95% confidence intervals (CI) are presented in the table below. Bold indicates a significant association.**

| microRNA | OR (95% CI) | AUROC (95% CI) |
|---|---|---|
| miR-10a-5p | 0.94 (0.57–1.56) | 0.55 (0.37–0.73) |
| miR-16-5p | 0.61 (0.35–1.05) | 0.63 (0.46–0.80) |
| miR-21-5p | **3.48 (1.27–9.53)** | **0.74 (0.58–0.89)** |
| miR-26b-5p | 1.01 (0.50–2.05) | 0.54 (0.37–0.72) |
| miR-27a-5p | 0.54 (0.23–1.26) | 0.63 (0.47–0.74) |
| miR-29a-5p | 1.11 (0.58–2.12) | 0.57 (0.39–0.74) |
| miR-93-3p | 0.84 (0.55–1.30) | 0.56 (0.35–0.78) |
| miR-101-3p | 0.79 (0.34–1.87) | 0.45 (0.28–0.63) |
| miR-127-3p | 1.00 (0.72 - 1.40) | 0.55 (0.34–0.76) |
| miR-146a-5p | 1.45 (0.73–2.89) | 0.59 (0.42–0.76) |
| miR-192-5p | 0.94 (0.63–1.39) | 0.56 (0.38–0.75) |
| miR-210-3p | 0.98 (0.65–1.48) | 0.52 (0.35–0.70) |

**Table 5. Summary of previous and current findings.**

**Knowledge before this study**

- miRNAs have been proposed as promising biomarkers for sepsis, kidney injury and SA-AKI in experimental and clinical studies.
- Both miR-21-5p and miR-16-5p are associated with kidney injury in murine sepsis models and are elevated in septic ICU patients, where they serve as predictors of mortality.
- Most prior studies on miRNAs in SA-AKI have been limited to animal models and comparisons with healthy controls, lacking validation in diverse, real-world clinical populations.

**What this study adds**

- The majority of miRNAs that had been associated with sepsis, kidney injury or SA-AKI in previous studies, could not be reproduced in this clinically relevant cohort of patients with sepsis, urging caution in their translational application.
- Increased plasma levels of miR-21-5p and decreased levels miR-16-5p are associated with early stage SA-AKI among patients at ED presentation.
- Only miR-21-5p consistently associates with SA-AKI, confirming its generalizability across relevant clinical settings.
- Plasma levels of miR-21-5p are independently associated with 30-day mortality among patients at the ED, but not among patients at the ICU.

These results align with prior studies that reported similar circulating miRNA patterns in patients with AKI [20,21,40], as well as in sepsis mouse models [40–43]. Experimental studies support the biological involvement of these miRNAs in kidney injury as they have been implicated in regulating inflammation, cellular stress responses, and apoptosis—key processes in the pathophysiology of SA-AKI [14,18,40,42,44–46]. Despite this, the diagnostic value of miR-21-5p and miR-16-5p in our cohort was modest: miR-21-5p achieved an AUC of 0.65—lower than the previously reported 0.83 in cardiac surgery-associated AKI [47]—and miR-16-5p yielded an AUC of 0.61 compared to 0.92 previously reported in paediatric ICU SA-AKI [40]. Combining both miRNAs did not enhance diagnostic performance. These findings suggest that although miR-21-5p and miR-16-5p are involved in key biological processes relevant to SA-AKI, their utility as standalone or combined early biomarkers at ED presentation remains limited.

The limited diagnostic value of miRNAs observed in our study reflects the complex, heterogeneous, and incompletely understood pathophysiology of SA-AKI, as well as the multifactorial and context-dependent nature of these miRNAs in this

condition. For example, the function of miR-21-5p in regulating inflammation is highly context- and cell type-dependent [17,18,46,48–50]. In lung endothelial cells, overexpression of miR-21-5p increased IL-6 and IL-1β levels, exacerbating inflammation in LPS induced acute lung injury [51]. Conversely, in tubular epithelial cells, miR-21-5p exhibited anti-inflammatory and cytoprotective effects after renal ischemia and reperfusion by inhibiting PDCD4/NF-κB, and protecting against apoptosis [19]. Thus, its role can be both pro- and anti-inflammatory depending on the cellular environment and disease context [17,18]. Moreover, neither miR-21-5p nor miR-16-5p are specific to kidney injury [40,52]. MiR-21-5p has been implicated in a wide range of conditions, including hepatic disorders [53,54], inflammatory lung disease [51], cardiovascular disease [55], and various cancers [56,57]. Similarly, miR-16-5p has been reported to be downregulated in the lungs and implicated in inflammatory responses during acute lung injury [40,58]. These findings suggest that circulating miRNAs may reflect inflammatory processes in general, rather than being able to serve as organ-specific biomarker. Therefore, a more comprehensive understanding of the biological roles of miR-21-5p and miR-16-5p is necessary to define their specific contributions to SA-AKI. Clarifying these mechanisms will be essential before miRNA-based diagnostics can be confidently integrated into clinical practice.

Despite selecting miRNAs through a rigorous literature review, supported by experimental evidence and prior validation to address common limitations, we were unable to replicate most previously reported associations with SA-AKI in our clinically relevant cohorts—neither at the early stage of hospital presentation nor during advanced disease in the ICU. One possible reason is the limited translatability of animal AKI models [59], which often rely on ischemic injury [60], whereas SA-AKI follows a different pathophysiology with milder histological changes [61]. For example, miR-192 upregulation has been reported in ischemia-reperfusion injury (IRI) models and in human AKI following cardiac surgery [60], which may not be generalizable to SA-AKI. Moreover, several miRNAs, such as miR-21, miR-29, and miR-146, have been linked to renal fibrosis and chronic kidney disease (CKD) [48,62,63]. For example, miR-21-5p is associated with TGF-β signalling in both human and animal models, and its blood and urine levels correlate with the extent of fibrosis [48,62,63]. In our cohort, patients with CKD were predominantly found in the ED SA-AKI and ICU groups, and chronic pathological changes may have influenced their circulating miRNA levels. Another possible explanation is that the systemic response in sepsis is biologically complex, with dynamic shifts between pro- and anti-inflammatory signals that are likely to influence miRNA expression over time. Opposing miRNA responses at different disease stages can effectively cancel each other out. This is supported by a review on miRNA expression in sepsis and infection, noting inconsistent up- or downregulation of miR-16, miR-21, and miR-146a across studies, likely reflecting their dynamic regulatory roles across the sepsis trajectory [64]. Accordingly, miRNA expression may vary depending on the timing of measurement relative to disease trajectory. We collected samples early at ED admission, whereas others focused on severe SA-AKI in the ICU [21]. To assess if timing differences accounted for the lack of replication, we studied an ICU cohort anticipating stronger signals but did not replicate the previously observed associations. Since the exact timing of the renal insult in SA-AKI is often unclear and AKI diagnosis can be delayed due to creatinine kinetics [39], miRNA measurements may not align with the true injury phase. Serial miRNA measurements have demonstrated that miR-192 levels fluctuate and can normalize within 24–72 hours post-injury [60], suggesting that miRNA differences might be missed if sampling occurs when renal function is already recovering. Another factor to consider is that these miRNAs may track global inflammatory and septic processes linked to severity, rather than specific organ involvement. This may explain the large variability seen in S6 Table for the ICU cohort. If these miRNAs reflect disease severity, they would lack discriminatory power since ICU sepsis and SA-AKI patients had comparable severity, reflected in similar APACHE scores, mortality, and length of stay. However, this does not explain why we found largely no association with mortality in either cohort (S6 Table). It is possible that previously reported associations between miRNAs and severity were overstated due to comparisons with healthy controls, which can exaggerate differences not seen in clinically relevant hospital populations. Finally, miRNA levels can be influenced by the differences of sample type. For instance, miR-10a and miR-21 were found to be elevated in kidney tissue following contrast-induced AKI in rat models, but this increase was not reflected in plasma levels in either rats or humans [65]. Taken together, factors

such as differences in AKI aetiology, timing of sample collection, sample types, and disease contexts—combined with contrasting whole body responses and frequent reliance on healthy controls—complicate cross-study comparisons and likely contribute to inconsistent findings. Our findings underscore the complexity of interpreting isolated molecular data within the broader whole-body pathophysiology of SA-AKI, highlighting challenges in translating miRNA biomarkers into clinical practice. This underscores the necessity of validation studies to determine their true diagnostic value and to advance their potential application in clinically relevant settings.

### Future implications

MiR-21-5p may serve as an early prognostic marker for sepsis patients in the ED. Our results showed that miRNA levels did not strongly correlate with conventional markers of inflammation, sepsis severity, or organ dysfunction. Therefore, circulating miRNAs may reflect an inflammatory or cellular stress response not apparent by clinical parameters such as abnormal vital parameters or increased lactate. This suggests their potential utility as complementary diagnostic tools, offering insights into immune dysregulation or tissue-specific injury mechanisms independent of classical inflammatory markers. For example, inhibition of miR-21-5p increases oxidative stress *in vitro* in tubular epithelial cells in an ischemia model [50]. Future studies should assess whether incorporating miR-21-5p levels into early risk stratification models enhances prognostic accuracy and informs clinical decision-making in sepsis-associated AKI.

Some studies reported stronger associations between miRNAs and clinical outcomes when measured in urine rather than plasma [47]. Unlike plasma, which can contain miRNAs derived from various tissues and systemic responses, urinary miRNAs may more accurately capture kidney-specific changes, thereby offering improved specificity for renal injury. Additionally, assessing miRNAs alongside conventional renal biomarkers, such as the tubular injury marker KIM-1, could help determine whether they offer added diagnostic value. Moreover, given the dynamic nature of miRNA expression, a single measurement at ED presentation may not capture their full diagnostic or prognostic potential. Longitudinal studies of urinary and plasma miRNAs could clarify their changes over time in relation to SA-AKI onset, progression, and resolution, improving our understanding of their value in predicting renal and clinical outcomes as dynamic biomarkers.

### Limitations

This study has several limitations. Despite the thorough literature review, the selection of miRNA biomarker candidates was not exhaustive, meaning that some potentially valuable miRNAs may have been overlooked. The study was powered for the primary aim, findings in the ICU group were exploratory. S6 Table indicates that effect sizes for most miRNAs were small, suggesting that non-significant results likely reflect a lack of strong clinically or biologically relevant associations, rather than insufficient statistical power. An exception may be miR-16-5p, which showed a trend that might reach significance with a larger sample (n = 50/group). The inability to include urine output as a variable limits the precision of AKI diagnosis—a common challenge in clinical studies where urine output is either not consistently recorded or poorly documented in clinical practice [66]. Furthermore, while many studies estimate a baseline creatinine using back-calculation methods (e.g., CKD-EPI, MDRD), we deliberately chose to work only with available, measured data to avoid introducing additional bias [67,68]. This decision may have led to an underestimation of AKI incidence in patients without documented baseline kidney function.

### Conclusion

In conclusion, we selected twelve miRNAs based on a literature review and demonstrate that, despite their previous associations with sepsis, kidney injury or SA-AKI, they have limited utility as biomarkers for early SA-AKI recognition in clinical settings. Only circulating miR-21-5p and miR-16-5p were associated with SA-AKI at initial presentation, with miR-21-5p also independently predicting 30-day mortality. To assess generalizability, we validated these findings in an ICU cohort

with more advanced disease, where only miR-21-5p remained associated with SA-AKI, but not with mortality. This study underscores the importance of validating miRNA biomarkers across different clinical contexts, and highlights both the potential and limitations of miR-21-5p as a biomarker for SA-AKI.

## Supporting information

**S1 Table. Overview of literature review and selection of candidate miRNAs.** This table summarizes all studies identified and reviewed during the literature search that informed the selection of candidate microRNAs (miRNAs) for further investigation. The miRNAs ultimately selected for this study are indicated in bold. Reference numbers correspond to the supplemental reference list provided in S File 1, S3 Table. Studies highlighted in red reported negative findings regarding miRNA association or relevance. The thematic focus of each study is color-coded as follows: yellow indicates a focus on kidney-related outcomes including acute kidney injury (AKI), blue denotes sepsis-focused studies, and green signifies studies specifically investigating sepsis-associated AKI (SA-AKI).
(XLSX)

**S2 Table. Reference list for literature review tables.** This supplemental table provides the complete list of references corresponding to the literature reviewed and summarized in S1 Table and SFile 3: S3 Table. Each reference is assigned a number used for cross-referencing within the table. The studies included span a range of topics relevant to miRNA expression in kidney injury, sepsis, and sepsis-associated acute kidney injury (SA-AKI). These references were used to guide the identification and selection of the 12 candidate miRNAs investigated in the current study.
(XLSX)

**S3 Table. Schematic overview of reviewed literature by study focus and sample material.** A schematic overview of all literature reviewed, categorized by the main focus of each study and the type of biological material analyzed. References correspond to those listed in S2 Table . Studies are sorted by the sample material used—blood, urine, or tissue/cell cultures—and color-coded according to both the study model and disease focus.
(XLSX)

**S1 Fig. Bar graph representing the number of references per microRNA.** The microRNAs are ordered in descending order based on their reference count. The values correspond to those listed in S1 Table and S2 Table.
(XLSX)

**S2 Fig. Venn diagram illustrating miRNA associations with Kidney Injury, Sepsis, and SA-AKI.** This Venn diagram illustrates the number of microRNAs associated with kidney injury (yellow), sepsis (blue), and SA-AKI (green).
(XLSX)

**S3 Fig. Missingness per microRNA and cohort.** Overview of rate of missing microRNA values per cohort.
(DOCX)

**S4 Table. Site of infection.** Overview of the infection sites observed at Emergency Department (ED) and Intensive Care Unit (ICU) admission across the four subgroups with an infection.
(DOCX)

**S5 Table. 30-day mortality associations in the ED cohort.** Association of circulating microRNA levels with 30-day mortality in the ED cohort. Odds ratio (OR), and area under the receiver operator curve (AUROC) with their respective 95% confidence intervals (CI) are presented in the table below. **Bold** indicates significant association.
(DOCX)

**S6 Table. 30-day mortality associations in the ICU cohort.** Association of circulating microRNA levels from plasma with 30-day mortality in the ICU cohort. Odds ratio (OR), and area under the receiver operator curve (AUROC) with their respective 95% confidence intervals (CI) are presented in the table below. **Bold** indicates significant association. (DOCX)

**S4 Fig. Scatterplots on correlation laboratory parameters with miR-21-5p. Scatterplots exploring the correlation of miR-21-5p with laboratory parameters including eGFR, urea, creatinine, CRP and lactate.** The upper row represents the ED cohort, while the lower row represents the ICU cohort. Correlation coefficients (r) and corresponding p values are shown. (DOCX)

**S5 Fig. Correlation plot microRNAs and clinical parameters.** Correlation plot including microRNAs, laboratory parameters of inflammation, sepsis and kidney function and acute kidney injury severity based on KDIGO criteria. eGFR – estimated glomerular filtration rate, CRP – C reactive protein, AKI – Acute Kidney Injury, KDIGO – kidney disease improving global outcome. Red indicated a more negative correlation, blue indicates a more positive correlation. Heatmap shows the results from the ED cohort. (DOCX)

**S7 Table. Post hoc power analysis for the measured miRNA level differences.** Effect sizes are reported as Cohens *f* for comparison of 3 groups (ED) and Cohens *d* for comparison of two groups (ICU). For better interpretation of non-significant findings 95% confidence intervals (CIs) are included to illustrate the range of plausible effect sizes. The estimated sample sizes required to achieve 80% power are shown in the last column. (DOCX)

**S1 File. MicroRNA datafile.**
(XLSX)

## Acknowledgments

We gratefully acknowledge the HEALICS and Acutelines study groups for their valuable contributions to data acquisition and data infrastructure. We extend our sincere thanks to the patients for their participation in this study. We thank professor G. Molema for her advice and expertise.

## Author contributions

**Conceptualization:** Tamar J. van der Aart, Matthijs Luxen, Jacqueline Koeze, Jan C. ter Maaten, Matijs van Meurs, Hjalmar R. Bouma.

**Data curation:** Tamar J. van der Aart, Matthijs Luxen, Matthias Hackl, Matijs van Meurs, Hjalmar R. Bouma.

**Formal analysis:** Tamar J. van der Aart, Matthijs Luxen, Matthias Hackl, Matijs van Meurs, Hjalmar R. Bouma.

**Funding acquisition:** Hjalmar R. Bouma.

**Investigation:** Tamar J. van der Aart, Matijs van Meurs, Hjalmar R. Bouma.

**Methodology:** Tamar J. van der Aart, Matthias Hackl, Matijs van Meurs, Hjalmar R. Bouma.

**Project administration:** Tamar J. van der Aart, Matthijs Luxen, Jan C. ter Maaten, Matijs van Meurs, Hjalmar R. Bouma.

**Resources:** Tamar J. van der Aart, Matthias Hackl, Jan C. ter Maaten, Matijs van Meurs, Hjalmar R. Bouma.

**Software:** Tamar J. van der Aart.

                                    

**Supervision:** Matthijs Luxen, Jacqueline Koeze, Marco van Londen, Jan C. ter Maaten, Matijs van Meurs, Hjalmar R. Bouma.

**Validation:** Tamar J. van der Aart.

**Visualization:** Tamar J. van der Aart.

**Writing – original draft:** Tamar J. van der Aart.

**Writing – review & editing:** Tamar J. van der Aart, Matthijs Luxen, Jacqueline Koeze, Marco van Londen, Jan C. ter Maaten, Matijs van Meurs, Hjalmar R. Bouma.

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
