## [Decision Letter · Decision Letter 0]

2 Jul 2025

PONE-D-25-28018Validation of plasma microRNAs as biomarkers in sepsis associated acute kidney injury upon first clinical presentation reveals limited diagnostic and prognostic performancePLOS ONE

Dear Dr. van der Aart,

Thank you for submitting your manuscript to PLOS ONE. After careful consideration, we feel that it has merit but does not fully meet PLOS ONE’s publication criteria as it currently stands. Therefore, we invite you to submit a revised version of the manuscript that addresses the points raised during the review process.

We look forward to receiving your revised manuscript.

Kind regards,

Keiko Hosohata, Ph.D.

Academic Editor

PLOS ONE

 [Acutelines is cofounded by the UMCG

TJ van der Aart is supported by an MD PhD grant from the University of Groningen]. 

Additional Editor Comments (if provided):

Reviewers' comments:

Reviewer's Responses to Questions

**Comments to the Author**

1. Is the manuscript technically sound, and do the data support the conclusions?

Reviewer #1: Yes

Reviewer #2: Partly

2. Has the statistical analysis been performed appropriately and rigorously? 

Reviewer #1: Yes

Reviewer #2: Yes

3. Have the authors made all data underlying the findings in their manuscript fully available?

Reviewer #1: Yes

Reviewer #2: Yes

4. Is the manuscript presented in an intelligible fashion and written in standard English?

Reviewer #1: Yes

Reviewer #2: Yes

5. Review Comments to the Author

Reviewer #1: The research team conducted a post-hoc analysis of 240 patients to explore the potential of 12 miRNAs for early recognition of SA-AKI and the generalizability of the findings to populations with more advanced diseases. The results only showed the significant association of miR-21-5p with SA-AKI and its 30-day mortality while the rest of miRNAs were not validated.

1. Abstract. Please spell out ED before using this abbreviation.

2. Line 111. Typo “The Biobank”

3. With a modest sample size, extensive missing data (line 203), and without adjusting for multiple testing (12 miRNAs being evaluated), please discuss the robustness and generalizability of the results.

4. It would be informative to report the missing rate for all miRNAs to be analyzed.

5. As replicability is low, it would be helpful to compare and contrast the studies from the literature and current studies. For example, are their missing rate similar to those reported in the literature? Similar study design? Similar target populations, and so on.

Reviewer #2: The study design integrates prospective data from ED and ICU cohorts, enabling direct comparison of miRNA biomarkers across different disease severities, which addresses a critical gap in prior research. Following points need to be addressed to improve the manuscript.

1.In the method part, authors should clarify whether the sample size was powered to detect miRNA-driven differences, especially for subgroup analyses. Besides, the relatively small sample size in the ICU cohort may limit the statistical power to detect subtle differences, please report the post-hoc power analyses to address the reliability of non-significant findings.

2.The exclusion of miR-127 due to missing data requires further clarification. Please report the percentage of missing values. And the potential impact of missing data on miRNA measurements should be discussed.

3.Notably, the ED cohort used KDIGO criteria for AKI, while the ICU cohort applied NICE guidelines. The authors should justify the comparability of results across these definitions, as differences in AKI staging may confound miRNA associations. Please note and explain this in the manuscript.

4.Suggest consider adding additional confounders (e.g., vasopressor use, mechanical ventilation) to multivariable models, as these may influence miRNA expression in sepsis.

5.To establish clinical relevance, it was suggested to directly compare miRNA performance with established markers (like serum creatinine, lactate, NGAL). This would clarify whether miR-21-5p/miR-16-5p offer incremental diagnostic value beyond conventional indicators.

6.There are differences in baseline characteristics and severity between the ED and ICU cohorts. Whether these variations might affect miRNA dynamics and generalizability of the findings should also be discussed.

7.It was observed that the non-infectious control group exhibited elevated baseline creatinine levels. Have the authors considered whether baseline renal function may confound the observed miRNA levels? It would be important to clarify whether baseline renal parameters were adjusted for in statistical models or discussed in the context of potential influences on miRNA expression.

8.In the discussion part, please elaborate on why most selected miRNAs failed to replicate prior associations. Is this due to sepsis stage, sample type, or biological redundancy? Link to broader literature on miRNA stability in systemic inflammation was suggested.

6. PLOS authors have the option to publish the peer review history of their article (what does this mean? ). If published, this will include your full peer review and any attached files.

**Do you want your identity to be public for this peer review?** For information about this choice, including consent withdrawal, please see our Privacy Policy .

Reviewer #1: No

Reviewer #2: **Yes: ** DING Ying

---

## [Author Response · Author response to Decision Letter 1]

9 Aug 2025

Response to reviewers

Reviewer #1

The research team conducted a post-hoc analysis of 240 patients to explore the potential of 12 miRNAs for early recognition of SA-AKI and the generalizability of the findings to populations with more advanced diseases. The results only showed the significant association of miR-21-5p with SA-AKI and its 30-day mortality while the rest of miRNAs were not validated.

1. Abstract. Please spell out ED before using this abbreviation.

Thank you for your attention. We have adjusted this in the manuscript.

2. Line 111. Typo “The Biobank”

We have adjusted this in the manuscript.

3. With a modest sample size, extensive missing data (line 203), and without adjusting for multiple testing (12 miRNAs being evaluated), please discuss the robustness and generalizability of the results.

We appreciate the reviewer’s comments. To address them comprehensively, we have divided the response into three parts corresponding to the specific concerns raised. We will then conclude with a summary comment on the robustness and generalizability of our findings.

Sample size calculation (a priori)

In accordance with institutional requirements and with data minimization principles (GDPR), we were required to justify patient data use with an a priori sample size calculation for the institutional ethical committee of the UMCG. Based on reported differences in circulating miRNA levels in SA-AKI in an emergency department population (effect size 0.6, α = 0.05, β = 0.1) (1)[Lin et al., 2019], we calculated a required sample size of 60 patients per group. This sample size formed the basis of our data access request for our primary analysis and was approved by the relevant data governance and ethics bodies.

We did not perform an a priori sample size calculation for the ICU population comparison. However, based on literature, “a priori” sample size estimates for similar comparisons range from as few as 3 to as many as 42 patients. This wide range illustrates how variable the results could have been, underscoring the marked heterogeneity of these cohorts and the inherent challenges in achieving consistent statistical power.

Lorenzen et al. 2011 CJASN: AKI vs non healthy controls (MI):

o Based on group means for miR-16 levels (0.1 vs. 1.3), standard deviations (0.01 and 0.7), effect size (Cohen’s d) = -4 , using α = 0.05 and power = 0.8, the required sample size per group was 5

o Based on group means for miR-210 (11.0 vs. 2.5), standard deviations (2.0 and 1.25), effect size (Cohen’s d) = -3.7, using α = 0.05 and power = 0.8, the required sample size per group was 3

Gaede et al. 2016 NTD: prediction of AKI development following cardiac surgery

o Based on group medians [IQR] for miR-21 0.27 [0.30 - 0.14] vs. 0.44 [0.75 – 0.25] effect size (Cohen’s d) = 0.6, using α = 0.05 and power = 0.8, the required sample size per group was 42

Aguado-Fraile et al. 2015 PloSone: AKI at the ICU and healthy controls

o Based on group medians [IQR] for miR-27a 9 [7.5-11] vs. 0.5 [-1 – 2.5] effect size (Cohen’s d) = 3.3, using α = 0.05 and power = 0.8, the require d sample size per group was 3

Caserta et al. 2016 Sci Rep: Sepsis from SIRS

o Based on group medians [IQR] for miR-192 -3 [-2.5- -4.5] vs. -4.5 [-3 – -5.75] effect size (Cohen’s d) = 0.8, using α = 0.05 and power = 0.8, the required sample size per group was 23

Ramachandran et al. 2013 Cli Chem: AKI at the ICU vs ICU patients without AKI:

o Based on group medians for miR-21 11.0 [7.5-14.5] vs. 1.25 [1-1.5] effect size (Cohen’s d) = 2.6, using α = 0.05 and power = 0.8, the required sample size per group was 4

Post-hoc sample size calculation

To confirm that the observed group differences were supported by sufficient statistical power given our sample size, we performed a post hoc power analysis in R using package pwr, for the reported positive findings (miR-10a-5p (ED), miR-16-5p (ED) and miR-21-5p (ED, ICU)). The post hoc analysis showed that with the observed effect sizes (Cohen’s f =0.25, 0.24 and 0.35) our sample size exceeded the number required to achieve 80% power at α = 0.05 (Rebuttal Table 1). These results confirm that our study was sufficiently powered to detect the observed differences among groups. Despite the slight shortfall in ICU group size (18 vs. 21), the large observed effect for miR-21-5p (Cohen’s d = 0.89) provides reasonable confidence in the validity of the result.

To address these points we have added the post hoc power analyses to the supplemental material (S6 Table) and refer to this in the discussion (Line 449).

Next, assessing the robustness of non-significant findings is inherently challenging because statistical tests and confidence intervals cannot definitively prove the absence of a difference. However, we can assess the robustness of our non-significant findings by evaluating whether the study was sufficiently powered to detect meaningful effects. To this end, we repored effect sizes - Cohen’s d or f corresponding with three and two groups - with their corresponding 95% confidence intervals (S Table 6). These intervals provide a range of plausible values for the true effect size. For several miRNAs, these intervals included 0 (Cohen’s d) or were centered near zero (Cohen’s f). This could mean that the data are too variable or the sample size too small to precisely estimate the effect. We also reported the sample sizes needed to detect meaningful differences for these miRNAs at our calculated effect size, which in several cases would be very large. Overall, the results reflect either a genuinely small or no effect, or insufficient power to detect a meaningful difference, but argue against missing any large, clinically relevant effects due to limited sample size.

Taken together, this indicates that any true differences are likely too small to be biologically or clinically meaningful, and supports the interpretation that our non-significant findings likely reflect a true absence of strong effects rather than a lack of statistical power. To address this, we have revised the phrasing in the abstract, results, and discussion to clearly indicate that the ICU findings are exploratory. Additionally, we have added a statement to the Limitations section (line 496) regarding the robustness and interpretability of our findings in the ICU cohort:

“The study was powered for the primary aim, while the findings in the ICU group were exploratory. S6 Table indicates that effect sizes for most miRNAs were small, suggesting that non-significant results likely reflect a lack of strong clinically or biologically relevant associations, rather than insufficient statistical power. An exception may be miR-16-5p, which showed a trend that might reach significance with a larger sample (n=50/group).”

Missing data

Our reference to “extensive missing data” (line 198) pertained specifically to the calculation of the global mean normalization factor involving miR-127, which was absent in 34% and 26% of patients in the ED and ICU populations, respectively. In hindsight, this characterization was an overstatement, and we have accordingly removed the term “extensive” to more accurately reflect the data quality. For transparency, we have included a detailed overview of missing data in the Supplemental material (S1 Figure).

Multiple testing

We did not apply multiple testing correction because the 12 miRNAs were selected a priori based on literature and prior experimental evidence, making this a hypothesis-driven rather than exploratory analysis. In this context, strict correction methods risk inflating type II error and overlooking biologically relevant signals.

Robustness and Generalizability

In summary, given the adequately powered sample size, limited missing data, and a well-justified approach to multiple testing correction, we are confident that our findings are robust and generalizable for the primary outcome. In response to reviewer feedback, we have removed advanced statistical comparisons for the ICU cohort; these analyses are now presented as exploratory in the supplemental materials. We believe these revisions strengthen the validity of our results, while acknowledging that further validation in larger and more diverse populations is warranted.

4. It would be informative to report the missing rate for all miRNAs to be analyzed.

We thank the reviewer for this suggestion. While this information was already available in the shared dataset, we now explicitly include an overview of missing data per miRNA in the Supplementary Materials as mentioned above.

5. As replicability is low, it would be helpful to compare and contrast the studies from the literature and current studies. For example, are their missing rate similar to those reported in the literature? Similar study design? Similar target populations, and so on.

We appreciate this suggestion. While we had briefly addressed this issue in the Discussion, we recognize that a more in-depth comparison with existing literature would be valuable. We have therefore expanded the Discussion section to include additional context and relevant considerations (see below) and added a remark on increasing our understanding of the temporal relation with sepsis in the Future implications (line 488).

We appreciate this suggestion. While we had briefly addressed this issue in the Discussion, we recognize that a more in-depth comparison with existing literature would be valuable. We have therefore expanded the Discussion section to include additional context and relevant considerations (see below) and added a remark on increasing our understanding of the temporal relation with sepsis in the Future implications (line 488).

Discussion

“Despite selecting miRNAs through a rigorous literature review, supported by experimental evidence and prior validation to address common limitations, we were unable to replicate most previously reported associations with SA-AKI in our clinically relevant cohorts—neither at the early stage of hospital presentation nor during advanced disease in the ICU. One possible reason is the limited translatability of animal AKI models (59), which often rely on ischemic injury (60), whereas SA-AKI follows a different pathophysiology with milder histological changes (61). For example, miR-192 upregulation has been reported in ischemia-reperfusion injury (IRI) models and in human AKI following cardiac surgery (60), which may not be generalizable to SA-AKI. Moreover, several miRNAs, such as miR-21, miR-29, and miR-146, have been linked to renal fibrosis and chronic kidney disease (CKD) (48,62,63). For example, miR-21-5p is associated with TGF-β signalling in both human and animal models, and its blood and urine levels correlate with the extent of fibrosis (48,62,63). In our cohort, patients with CKD were predominantly found in the ED SA-AKI and ICU groups, and chronic pathological changes may have influenced their circulating miRNA levels. Another possible explanation is that the systemic response in sepsis is biologically complex, with dynamic shifts between pro- and anti-inflammatory signals that are likely to influence miRNA expression over time. Opposing miRNA responses at different disease stages can effectively cancel each other out. This is supported by a review on miRNA expression in sepsis and infection, noting inconsistent up- or downregulation of miR-16, miR-21, and miR-146a across studies, likely reflecting their dynamic regulatory roles across the sepsis trajectory (64). Accordingly, miRNA expression may vary depending on the timing of measurement relative to disease trajectory. We collected samples early at ED admission, whereas others focused on severe SA-AKI in the ICU (21). To assess if timing differences accounted for the lack of replication, we studied an ICU cohort anticipating stronger signals but did not replicate the previously observed associations. Since the exact timing of the renal insult in SA-AKI is often unclear and AKI diagnosis can be delayed due to creatinine kinetics (39), miRNA measurements may not align with the true injury phase. Serial miRNA measurements have demonstrated that miR-192 levels fluctuate and can normalize within 24–72 hours post-injury (60), suggesting that miRNA differences might be missed if sampling occurs when renal function is already recovering. Another factor to consider is that these miRNAs may track global inflammatory and septic processes linked to severity, rather than specific organ involvement. This may explain the large variability seen in S6 Table for the ICU cohort. If these miRNAs reflect disease severity, they would lack discriminatory power since ICU sepsis and SA-AKI patients had comparable severity, reflected in similar APACHE scores, mortality, and length of stay. However, this does not explain why we found largely no association with mortality in either cohort (S6 Table). It is possible that previously reported associations between miRNAs and severity were overstated due to comparisons with healthy controls, which can exaggerate differences not seen in clinically relevant hospital populations. Finally, miRNA levels can be influenced by the differences of sample type. For instance, miR-10a and miR-21 were found to be elevated in kidney tissue following contrast-induced AKI in rat models, but this increase was not reflected in plasma levels in either rats or humans (65). Taken together, factors such as differences in AKI aetiology, timing of sample collection, sample types, and disease contexts—combined with contrasting whole body responses and frequent reliance on healthy controls—complicate cross-study comparisons and likely contribute to inconsistent findings. Our findings underscore the complexity of interpreting isolated molecular data within the broader whole-body pathophysiology of SA-AKI, highlighting challenges in translating miRNA biomarkers into clinical practice. This underscores the necessity of validation studies to determine their true diagnostic value and to advance their potential application in clinically relevant settings.”

Future implications

“ Longitudinal studies of urinary and plasma miRNAs could clarify their changes over time in relation to SA-AKI onset, progression, and resolution, improving our understanding of their value in predicting renal and clinical outcomes as dynamic biomarkers. ”

Reviewer #2

The study design integrates prospective data from ED and ICU cohorts, enabling direct comparison of miRNA biomarkers across different disease severities, which addresses a critical gap in prior research. Following points need to be addressed to improve the manuscript.

1.In the method part, authors should clarify whether the sample size was powered to detect miRNA-driven differences, especially for subgroup analyses. Besides, the relatively small sample size in the ICU cohort may limit the statistical power to detect subtle differences, please report the post-hoc power analyses to address the reliability of non-significant findings.

We appreciate the reviewer’s comment and have thoroughly addressed it in our response to reviewer #1 question 3. As a result, S6 Table has been added to the supplementary materials, the discussion has been updated accordingly (see response Q3) and a relevant statement has been included in the Limitations section (line 496).

2.The exclusion of miR-127 due to missing data requires further clarification. Please report the percentage of missing values. And the potential impact of missing data on miRNA measurements should be discussed.

Our reference to “extensive missing data” (line 198) pertained specifically to the calculation of the global mean normalization factor involving miR-127, which was absent in 34% and 26% of patients in the ED and ICU populations, respectively. In hindsight, this characterization was an overstatement, and we have accordingly removed the term “extensive” to more accurately reflect the data quality. For transparency, we have included a detailed overview of missing data in the Supplemental material (S1 Figure).

3.Notably, the ED cohort used KDIGO criteria for AKI, while the ICU cohort applied NICE guidelines. The authors should justify the comparability of results across these definitions, as differences in A

---

## [Editor Report · Decision Letter 1]

17 Aug 2025

Validation of plasma microRNAs as biomarkers in sepsis associated acute kidney injury upon first clinical presentation reveals limited diagnostic and prognostic performance

PONE-D-25-28018R1

Dear Dr. van der Aart,

We’re pleased to inform you that your manuscript has been judged scientifically suitable for publication and will be formally accepted for publication once it meets all outstanding technical requirements.

Kind regards,

Keiko Hosohata, Ph.D.

Academic Editor

PLOS ONE
---

## [Editor Report · Acceptance letter]

PONE-D-25-28018R1

PLOS ONE

Dear Dr. van der Aart,

I'm pleased to inform you that your manuscript has been deemed suitable for publication in PLOS ONE. Congratulations! Your manuscript is now being handed over to our production team.

Kind regards,

on behalf of

Dr Keiko Hosohata

Academic Editor

PLOS ONE